# Diet in Patients with Myocardial Infarction and Coexisting Type 2 Diabetes Mellitus

**DOI:** 10.3390/ijerph20085442

**Published:** 2023-04-07

**Authors:** Elżbieta Szczepańska, Magdalena Gacal, Adam Sokal, Barbara Janota, Oskar Kowalski

**Affiliations:** 1Department of Human Nutrition, Department of Dietetics, Faculty of Public Health in Bytom, Medical University of Silesia in Katowice, Jordana 19 Street, 41-808 Zabrze, Poland; 2Doctoral School, Faculty of Public Health in Bytom, Medical University of Silesia in Katowice, Piekarska 18 Street, 41-902 Bytom, Poland; 31st Department of Cardiology, Silesian Center for Heart Diseases, Marii Curie-Skłodowskiej 9 Street, 41-800 Zabrze, Poland; 4Department of Basic Medical Sciences, Faculty of Public Health in Bytom, Doctoral School of Medical University of Silesia in Katowice, Piekarska 18 Street, 41-902 Bytom, Poland

**Keywords:** dietary prevention, coronary artery diseases, diabetes mellitus, dietary deviations, dietary awareness

## Abstract

Background: Dietary modifications are recommended alongside pharmacotherapy in treating both diabetes mellitus (DM) and coronary heart disease (CHD) patients. Aims: The primary aim of our study was to assess the diet in patients with type 2 diabetes mellitus (T2DM) and myocardial infarction (MI) and to identify dietary differences between patients after the first and subsequent cardiovascular (CV) event. The secondary aim was to analyze the differences between men’s and women’s diets. Methods: The study population consisted of patients with DM/T2DM and MI. The research tool was the original author’s questionnaire which was collected personally by a qualified dietician. Results: The study included 67 patients with a mean age of 69 ± 8 years, hospitalized at the Silesian Centre for Heart Diseases in Zabrze in 2019. The study found that patients consumed less bread, whole-grain cereal products, fermented milk products, and vegetables than was recommended. A total of 32.8% of patients reported an intake of sweetened beverages, while 85.1% of participants consumed sweets despite being diagnosed with DM. Except for sweetened drinks, no differences in dietary behaviors were found in the patients after the first and second MI episode. Most of the included patients assessed their diet as appropriate. Conclusion: The dietary assessment of diabetes and myocardial infarction patients indicates that the diet does not comply with dietary recommendations, thus increasing the risk of a recurrent cardiac event despite a previous MI. No differences between the men’s and the women’s nutritional habits were observed.

## 1. Introduction

Ischemic heart disease (IHD) is one of the leading causes of death worldwide, with DM being a significant risk factor [1,2]. Both diabetes and impaired glucose tolerance (prediabetes) significantly increase the risk of MI in men and women compared to patients without DM [3]. The risk of MI associated with DM is more significant in women than men [4]. Disorders that increase CV risk, such as dyslipidemia, high blood pressure, or obesity, are more common in patients with DM and increase the risk of MI [2]. Long-term diabetes can lead to diabetic cardiomyopathy (DCM) [5,6]. Patients with MI and coexisting DM have an increased risk of CV events and a reduced life expectancy of about 7.5–20 years [7,8]. DM is often diagnosed only after CV incidents, such as MI [9].

Adequate nutrition and physical exercise are essential in preventing DM and CV diseases (CVDs) and their complications once the disorder is present. The current recommendations for diabetes mellitus and the prevention of CVDs point to the need for lifestyle modification, physical activity, and dietary modifications as the first critical steps taken before or simultaneously with pharmacotherapy [2,10]. Necessary dietary changes include limited calorie intake, reduced consumption of sodium and saturated fatty acids, including trans fatty acids, increased intake of vegetables and fruits, limited alcohol intake, smoking cessation, and increased physical activity. Saturated fatty acids should account for no more than 10% of total dietary energy in non-dyslipidaemic individuals and should not exceed 7% in those with dyslipidemia. Furthermore, it is also recommended that oily sea fish, a source of omega-3 fatty acids, be consumed at least twice a week. As for carbohydrates, complex sugars rich in dietary fiber should dominate, whereas simple sugars (mono- and disaccharides) should not exceed 10% of the energy supply [2,10,11].

The Mediterranean diet, which regulates glucose and triglycerides and normalizes HDL cholesterol, is one of the recommended dietary models for the population of MI patients with coexisting T2DM [12]. The Mediterranean diet is based on a high consumption of vegetables, fruit, fish, grains, olive oil, nuts, and a low intake of red wine, dairy products, red meat, cream, and sugary beverages. It is characterized by a low content of saturated fat and a high content of monounsaturated fat [12]. The recently published CARDIOPREV study confirmed its efficacy as a secondary prevention in reducing CV risk [13]. Additionally, it was shown that women are more prone to leading healthy lifestyles than men [14]. Therefore, one can expect that in the case of high-CV-risk patients, there would be differences in nutritional behaviors between individuals with and without a history of former CV episodes, and between men and women. Nonetheless, this question has not been sufficiently studied to date.

The primary aim of our study was to assess the qualitative and semiquantitative features of the diet of patients with T2DM and MI and to identify the qualitative and semiquantitative dietary differences between patients after the first and subsequent CV event. Our secondary aim was to analyze the qualitative and semiquantitative differences between men’s and women’s diets.

## 2. Materials and Methods

### 2.1. Study Population

The study was conducted following the principles of The Declaration of Helsinki. The Medical University of Silesia Bioethical Committee in Katowice approved the study protocol (Resolution No. KNW/0022/KB1/92/18 of 20 November 2018). Consecutive patients with DM, who had been hospitalized for myocardial infarction in the 1st Department of Cardiology of the Silesian Centre for Heart Diseases between 1 January and 31 December 2019, were invited to participate in the study. The inclusion criteria were: (1) age over 18 years, (2) diagnosed with T2DM, (3) past MI, and (4) informed consent to participate in the study. The exclusion criteria were (1) complicated MI or (2) lack of consent to participate in the study. Eligible patients were informed about the research procedures and gave their informed consent to participate in the study.

### 2.2. Research Tools and Procedures

The research tool was the original author’s questionnaire, consisting of closed and open-ended questions. The questionnaire consisted of 32 questions and enquired about personal data (gender, body weight, height, family situation), comorbidities, and lifestyle, including eating behaviors and the frequency of consumption of selected food products, physical activity, and alcohol before MI. Answers to 18 of these questions have been included in the current analysis. Answers to non-nutritional questions have been presented in Appendix A. The entire study questionnaire was attached to the auxiliary data. A certified dietitian conducted the survey. Anthropometric measurements, including body weight and height, were then used to calculate the participants’ body mass index (BMI). According to the WHO criteria, a BMI ≥ 25 kg/m^2^ was considered overweight, while obesity was defined as BMI ≥ 30 kg/m^2^.

### 2.3. Statistical Analysis

Statistical analysis was performed using Statistica version 13.3 software (TIBCO Software Inc., Palo Alto, CA, USA). Measurable data were presented as mean ± standard deviation. Categorical values were presented as numbers and percentages (Fraction; %). The two-tailed Student’s *t* test was used to compare the measurable data once the Kolmogorov–Smirnov test confirmed the normality of sample distribution. The test for two fractions was used to compare the percentage values. Significance was set at *p* < 0.05.

## 3. Results

The study included 67 patients aged 43–86 years (mean 69 ± 8 years), including 28 (41.8%) women and 39 (58.2%) men with a history of myocardial infarction with coexisting type 2 diabetes, hospitalized at the Silesian Centre for Heart Diseases in Zabrze in 2019. There were 15 (22.4%) participants residing in cities with more than 100,000 inhabitants and 47 (70.1%) participants living in towns with ≤100,000 inhabitants. There were 5 (7.5%) rural respondents. We included 33 (49.3%) and 34 (50.7%) patients after the first and second MI episodes, respectively. A total of 36 (53.5%) patients reported relevant cardiovascular family history (e.g., in parents or siblings).

Most of the study participants were overweight (n = 47; 70.1%). Obesity and overweight was found in 21 (63.6%) and 26 (76.5%) participants after the first and second MI episode, respectively (*p* = 0.378 first vs. second MI episode). Obesity and overweight were found in 19 (67.9%) women and 28 (71.8%) (*p* = 0.939) men. The mean BMI for the group was 28.2 ± 5.1, 28.5 ± 5.1 (mean ± SD) for those after their first MI episode, 27.9 ± 5.1 (mean ± SD); (*p* = 0.598) after the second MI episode, 26.6 ± 5.7 (mean ± SD) in women, and 27.9 ± 4.7 (mean ± SD) (*p* = 0.603) in men (Table 1). All participants had been diagnosed with type 2 DM. Declared (based on medical history) average longevity of DM was 7.04 ± 2.74 years (mean ± SD). The average HbA1C level (the most recent to the date of the survey) was 7.26 ± 1.29%. The incidence of comorbidities is shown in Table 2.

The analysis of our findings showed that most patients in the study consumed bread and whole-grain cereal products, fermented milk products, and vegetables less often than they should have. The percentage of patients consuming these products daily and having at least 3–4 servings of vegetables per day was 41.8%, 6%, 40.3%, and 23.9%, respectively. No differences were found in the frequency of consumption of these products between patients after the first vs. second MI episode (except for the intake of whole-grain cereal products) or between women and men (Table 3 and Table 4).

An analysis of the consumption rates for products not recommended in the daily diet of our patients showed a rare intake of red meat, and almost no information on fast foods or ready-to-heat meals, i.e., 67.2%, 75.2%, and 67.1%, respectively. Further analyses showed that some patients reported an intake of sweetened beverages (32.8%) and that a large group of participants consumed sweets (85.1%) despite being diagnosed with DM. No differences were found in the consumption rates for these products between patients after the first vs. second MI episode or between women and men (Table 5 and Table 6).

Furthermore, the study showed that only 52.2% of participants consumed adequate fluids. A total of 34.3% of patients took sugar in their beverages, and 73.1% added salt to cooked meals, which was alarming. No differences were found in the frequency of these behaviors between patients after the first vs. second MI episode (except for sweetening beverages) or between women and men (Table 7 and Table 8).

The respondents were also asked to self-report their diets. It was found that up to 53 (79.1%) patients, including 24 (72.7%) after their first MI and 29 (85.3%) after their second, as well as 22 (78.6%) women and 31 (79.5%) men considered their diets to be appropriate despite making multiple inappropriate dietary choices (Figure 1 and Figure 2).

## 4. Discussion

The major finding of this study was that the adherence of high-CV-risk patients with T2DM to dietary recommendations was poor, even in individuals who had previously experienced CV events. Additionally, contrary to expectations, the women’s nutritional habits were no different from the men’s. This study found that high-risk people, such as patients with T2DM, are unaware of current dietary recommendations and do not apply them even if they have experienced previous CV events; this observation provides new and important information which emphasizes the need for more extensive educational and motivational programs to be addressed to these patients. To the best of our knowledge, the direct comparison of nutritional behaviors between high-CV-risk patients with T2DM and CHD with and without previous CV events has yet to be reported. Therefore, this study provides new insight into the lifestyle implications of CV prevention.

This work is focused on nutritional behaviors. We know that complex lifestyle assessments also comprise other factors that were not included in the current analysis to maintain work integrity. Data regarding different aspects of lifestyle were included as a Appendix A. Proper nutrition and an active lifestyle are critical elements in preventing cardiovascular diseases. Following a healthy diet, maintaining an appropriate body weight, avoiding smoking, and practicing sports are crucial, though often neglected, factors in the secondary prevention of MI [10,15]. Similarly, nutritional interventions and lifestyle modification should occur before or simultaneously with pharmacotherapy to prevent and treat T2DM [2,16]. It has been shown that patients’ knowledge of preventing ischemic heart disease (IHD) is insufficient. They often lack knowledge of the risk factors and the principles of a proper diet [17]. Similar conclusions can be drawn from our study. Despite many poor dietary choices made by patients, up to 71.1% of them self-reported their diet as appropriate. This may indicate a low awareness of prevention and proper nutrition principles in CVDs and DM.

Patients diagnosed with heart failure are recommended to follow a Mediterranean diet with proven beneficial health effects and secondary prevention of CVDs [13]. This diet is also recommended for the prevention of diabetes [12]. This nutrition model is characterized, among other features, by a high fiber content from whole grains, vegetables, and fruits. As shown in our research, the frequency of consumption of fiber products is, however, insufficient. It has been shown that consuming high-fiber foods by patients with CAD after coronary intervention reduces the risk of future CV events by 38% [18]. The risk of IHD decreases with an increased total fiber consumption and fruit and vegetables combined [19]. Consuming fewer than three servings of vegetables and fruit per day was associated with a significantly higher risk of CV events with an increase of 2.22% (95% CI, 1.06–4.66), compared to those who consumed more than three servings of these products per day [20]. Our results have shown that patients with DM have an inappropriate diet, which does not change even in the case of a previous MI episode. At the same time, most of the surveyed patients self-reported their diet as being appropriate, indicating that the main reason these issues was a lack of basic knowledge about the dietary prevention of CVDs and DM. A statistically significant reduction in the percentage of patients sweetening beverages and a higher rate of those declaring the intake of products rich in complex carbohydrates (e.g., groats or rice) were the only effects reported in the group of patients after MI who should be motivated to use dietary prevention. Additional nutritional restrictions in this group of patients should apply not only to simple carbohydrates in the production and preparation of dishes but also to free sugars, primarily in sweets, sweetened beverages, and juices [2]. In our study, the consumption of sugar, sweetened beverages, and sweets was found to be too frequent. Surprisingly, both patients after the first MI episode and those with previous episodes saw no need to limit their intake of sweets and sweetened beverages. Furthermore, although the results of other authors suggested that women consumed more vegetables and fruits than men [19], we did not observe a significantly higher consumption of these products by women in our sample.

In addition to the high content of whole-grain products and vegetables, it is recommended to consume polyunsaturated fatty acids and fish, mainly marine, due to the presence of omega-3 fatty acids [10,12]. Many studies indicate a beneficial effect of omega-3 fatty acids on CV events [21,22,23]. Furthermore, a relationship was demonstrated to exist between the plasma levels of polyunsaturated fatty acids and CV risk. Among 944 MI patients, those with higher plasma levels of eicosatetraenoic acid (EPA) and plant α-linolenic acid (ALA) had a lower chance of suffering another acute CV event [24]. In our population, 86.6% of patients consumed fish with the recommended frequency, i.e., at least 1–2 times a week, including 84.8% after the first MI event and 88.2% after the second.

In addition to the proper amount of fish in the diet, our results indicate that red meat and fast food products are (although infrequently) present in the diet of patients. Many studies have shown that the limited intake of saturated fats, also present in red meat, contributed to a reduced cardiovascular risk [25,26,27]. The products mentioned above and fast foods are also natural sources of sodium. Consuming these, combined with adding salt to meals at the table, significantly increases the risk of hypertension. In our study, 73.1% of patients considered adding salt to already cooked meals unfavorable in the case of CVDs. This percentage is similar to that found in the study of Kucharska K et al., where only 37% of study participants declared dietary salt restriction [18]. Our study showed poor compliance with the declared eating habits in the population of DM patients after MI compared to the documented and confirmed results of clinical trials for dietary recommendations in this group of patients. A significant discrepancy between the presence of obviously inappropriate dietary choices and the self-reported appropriateness of dietary behaviors indicates this group of patients’ lack of elementary nutritional knowledge. This necessitates local- and large-scale educational programs, especially for patients with a high cardiovascular risk.

## 5. Conclusions

The diet assessment of diabetes and myocardial infarction patients indicates that they make poor dietary choices, increasing the risk of recurrent cardiovascular events. A history of MI has a minor impact on dietary modification, and nutritional mistakes are made similarly by both women and men. This highlights the need for providing high-CV-risk patients, especially patients with T2DM, with more appropriate dietary counseling.

### Study Limitations

The limitations of our research may have resulted from the small sample size, which reduced the study’s statistical power.

Another potential limitation is that only nutritional factors were included in the analysis, omitting such elements as smoking, alcohol consumption, and physical activity. Due to current guidelines, complex lifestyle modification is necessary to reduce the risk of CV disorders. However, to maintain the study’s integrity, we did not include other factors and only concentrated on the diet in this analysis.

## Figures and Tables

**Figure 1 ijerph-20-05442-f001:**
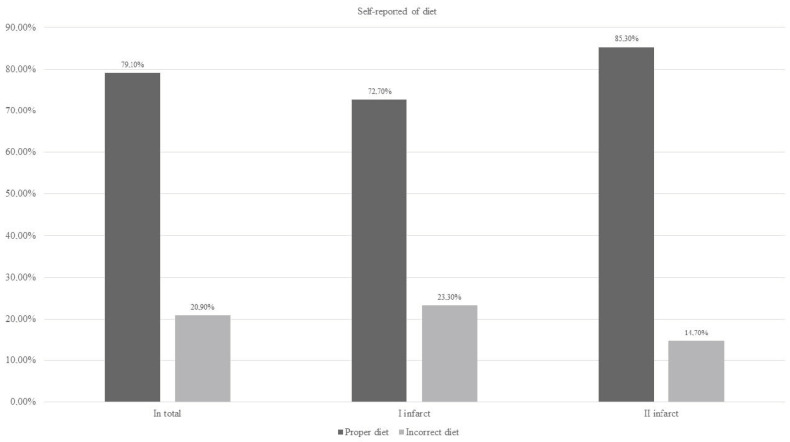
Self-reported diet in the study group of patients in total and by MI history (n = 67).

**Figure 2 ijerph-20-05442-f002:**
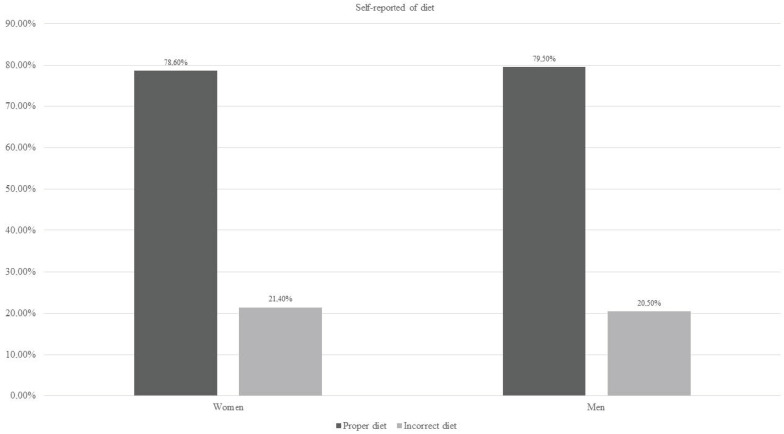
Self-reported diet in the study group of patients by gender (n = 67).

**Table 1 ijerph-20-05442-t001:** BMI in the study group (n = 67).

BMI	Total	First MI	Second MI	Women	Men
	n = 67	n = 33	n = 34	n = 28	n = 39
obesity	21 (31.3%)	14 (42.4%)	12 (35.3%)	12 (42.9%)	14 (35.9%)
overweight	26 (38.8%)	7 (21.2%)	14 (41.2%)	7 (25.0%)	14 (35.9%)
normal body weight	20 (29.9%)	12 (35.4%)	8 (23.5%)	9 (32.1%)	11 (28.2%)
mean BMI	28.2 ± 5.1	28.5 ± 5.1	27.9 ± 5.1	26.6 ± 5.7	27.9 ± 4.7

**Table 2 ijerph-20-05442-t002:** Comorbidities in the study group (n = 67). * more than one answer could be chosen.

Patients’ Diseases *	Total	First MI	Second MI	Women	Men
	n = 67	n = 33	n = 34	n = 28	n = 39
kidney failure	27 (40.3%)	13 (39.4%)	14 (41.2%)	8 (11.9%)	19 (28.4%)
thyroid disease	17 (25.4%)	7 (21.2%)	10 (29.4%)	9 (13.4%)	8 (11.9%)
atherosclerosis	24 (35.8%)	9 (27.3%)	15 (44.1%)	11 (16.4%)	13 (19.4%)
anemia	7 (10.4%)	5 (15.2%)	2 (5.9%)	3 (4.5%)	4 (6%)
liver failure	9 (13.4%)	4 (12.1%)	5 (14.7%)	4 (6%)	5 (7.5%)

**Table 3 ijerph-20-05442-t003:** The frequency of consuming food products recommended in the study group of patients, in total and by MI history (n = 67). * 65 people answered this question.

Frequency of Consumption		Total		First MI		Second MI		*p*
		n = 67	%	n = 33	%	n = 34	%	
Whole wheat bread	daily	28	41.8	10	30.3	18	52.9	0.056
	less often	32	47.8	18	54.5	14	41.2	0.25
	not at all	7	10.4	5	15.1	2	5.9	0.29
Whole-grain groats, rice, pasta	daily	4	6	3	9	1	2.9	0.30
	less often	58	86.7	25	75.8	33	97.1	0.0115
	not at all	5	7.5	5	15.2	0	0	0.0189
Fermented milk drinks	daily	27	40.3	10	30.3	17	50.0	0.09
	less often	38	56.7	22	66.7	16	47.1	0.09
	not at all	2	3.0	1	3.0	1	2.9	1.0
Fish	at least 1–2 portions a week	58	86.6	28	84.8	30	88.2	0.72
	less often	6	8.9	2	6.1	4	11.8	0.39
	not at all	3	4.4	3	9.1	0	0	0.08
Vegetables	at least 3–4 portions a day	16	23.9	10	30.3	6	17.7	0.25
	less often	46	68.7	21	63.6	26	76.5	0.63
	not at all	5	7.5	2	6.1	2	5.8	0.64
Fruit	at least 1–2 portions a day	52	77.6	26	78.8	26	76.5	0.64
	less often	13	19.4	7	21.2	6	17.6	0.76
	not at all	2	3.0	0	0	2	5.9	0.15
The form of consumed fruit *	raw	52	80.0	24	72.7	28	87.5	0.13
	juices in a carton or in a bottle	1	1.5	1	3.0	0	0	0.31
	boiled	12	18.5	8	24.3	4	12.5	0.2

**Table 4 ijerph-20-05442-t004:** The frequency of consuming food products recommended in the study group of patients by gender (n = 67). * 65 people answered this question.

The Frequency of Consumption		Women		Men		*p*
		n = 28	%	n = 39	%	
Whole wheat bread	daily	12	42.3	16	41.0	0.92
	less often	15	53.6	17	43.6	0.61
	not at all	1	3.5	6	15.4	0.38
Whole-grain groats, rice, pasta	daily	2	7.2	2	5.1	0.63
	less often	24	85.6	34	87.1	0.24
	not at all	2	7.2	3	7.7	0.35
Fermented milk drinks	daily	12	42.9	15	38.5	0.68
	less often	16	57.1	22	56.4	1.0
	not at all	0	0	2	5.1	0.23
Fish	at least 1–2 portions a week	24	85.7	34	87.2	0.91
	less often	4	14.3	2	5.1	0.13
	not at all	0	0	3	7.7	0.13
Vegetables	at least 3–4 portions a day	7	25.0	9	23.1	0.85
	less often	20	71.4	26	66.7	0.73
	not at all	1	3.6	4	10.2	0.36
Fruit	at least 1–2 portions a day	23	82.1	29	74.4	0.44
	less often	4	14.3	9	23.0	0.36
	not at all	1	3.6	1	2.6	0.82
The form of consumed fruit *	raw	19	70.4	33	86.8	0.09
	juices in a carton or in a bottle	1	3.7	0	0	0.21
	boiled	7	25.9	5	13.2	0.44

**Table 5 ijerph-20-05442-t005:** Consumption rates for food products not recommended in the study group of patients in total and by MI history (n = 67).

The Frequency of Consumption		Total		First MI		Second MI		*p*
		n = 67	%	n = 33	%	n = 34	%	
Red meat	daily	21	31.3	13	39.4	8	23.5	0.19
	less often	45	67.2	19	57.6	26	76.5	0.12
	not at all	1	1.5	1	3.0	0	0	0.31
Fast-food dishes	daily	0	0	0	0	0	0	1.00
	less often	16	23.8	10	30.3	6	17.6	0.25
	not at all	51	75.2	23	69.9	28	82.4	0.25
Ready to eat dishes	daily	0	0	0	0	0	0	0
	less often	22	32.9	13	39.0	9	26.5	0.26
	not at all	45	67.1	20	61.0	25	73.5	0.26
Sweetened carbonated beverages	daily	8	11.9	4	12.1	4	11.7	1.00
	less often	14	20.9	7	21.2	7	20.1	0.91
	not at all	45	67.2	22	66.7	23	67.6	0.91
Sweets	daily	6	9	4	12.1	2	5.9	0.40
	less often	51	76.1	26	78.8	25	73.5	0.57
	not at all	10	14.9	3	9.1	7	20.6	0.17

**Table 6 ijerph-20-05442-t006:** Consumption rates for food products not recommended in the study group of patients by gender (n = 67).

The Frequency of Consumption		Women		Men		*p*
		n = 28	%	n = 39	%	
Red meat	daily	7	25.0	14	35.9	0.34
	less often	20	71.4	25	64.1	0.49
	not at all	1	3.6	0	0	0.21
Fast-food dishes	daily	0	0	0	0	1.00
	less often	9	32.1	7	17.9	0.18
	not at all	19	67.9	32	82.1	0.18
Ready to eat dishes	daily	0	0	0	0	1.00
	less often	8	28.6	14	35.9	0.54
	not at all	20	71.4	25	64.1	0.54
Sweetened carbonated beverages	daily	2	7.1	6	15.4	0.48
	less often	5	17.9	9	23.1	0.76
	not at all	21	75.0	24	61.5	0.47
Sweets	daily	2	7.1	4	10.3	0.67
	less often	23	82.2	28	72.8	0.34
	not at all	3	10.7	7	17.9	0.43

**Table 7 ijerph-20-05442-t007:** Selected dietary behaviors in the study group in total and by MI history (n = 67). * except milk.

Behavior		Total		First MI		Second MI		*p*
		n = 67	%	n = 33	%	n = 34	%	
Number of consumed drinks *	5–6 glasses a day	35	52.7	17	51.5	18	52.9	0.94
	less	32	47.8	16	48.5	16	47.1	0.94
Sweetening drinks	yes	23	34.3	18	54.5	5	14.7	0.0006
	no	44	65.7	15	45.5	29	85.3	0.0006
Salting dishes	yes	49	73.1	23	69.7	26	76.5	0.58
	no	18	26.9	10	30.3	8	23.5	0.58

* is number of consumed drinks.

**Table 8 ijerph-20-05442-t008:** Selected dietary behaviors in the study group by gender (n = 67). * except milk.

Behavior		Women		Men		*p*
		n = 28	%	n = 39	%	
Number of consumed drinks *	5–6 glasses a day	13	46.4	22	56.4	0.75
	less	15	53.6	17	43.6	0.75
Sweetening drinks	yes	12	42.9	11	28.2	0.20
	no	16	56.1	28	71.8	0.20
Salting dishes	yes	22	78.6	27	69.2	0.86
	no	6	21.4	12	30.8	0.72

* is number of consumed drinks.

## Data Availability

The source data are available on request.

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
