# Peer review of "Diet in Patients with Myocardial Infarction and Coexisting Type 2 Diabetes Mellitus"

_ijerph, 2023, doi:10.3390/ijerph20085442_

Round 1

Reviewer 1 Report

1. Since respondents were diabetics, are they all Type 2 DM? It will be informative to table data on the diabetes patients, say how long have they been diabetic and their blood glucose & HbA1C levels.

2. In the Introduction, first paragraph, second sentence, line38, there is a missing word - significant in women ? in men.

3. Abstract - Conclusion - Incomplete sentence. What is ca?

Author Response

Thank you for your comments; I have included the replies to detailed questions below. Corrections in the text of the manuscript were flagged in red.

1. Since respondents were diabetics, are they all Type 2 DM? It will be informative to table data on the diabetes patients, say how long have they been diabetic and their blood glucose & HbA1C levels.

RE:  Thank you for this comment. All respondents were Type 2 DM. Declared (based on medical history) average longevity of DM was 7.04 ± 2.74  years (mean ± SD).  The average HbA1C level (the most recent to the date of the survey) was 7,26 ± 1,29%) These data were included in the manuscript.

2. In the Introduction, first paragraph, second sentence, line38, there is a missing word - significant in women ? in men.

RE  Sorry for this typo. This was corrected in the text.

3. Abstract - Conclusion - Incomplete sentence. What is ca?

RE  Sorry for this typo. This was corrected in the text.

Reviewer 2 Report

In order to assess the diet of type 2 diabetes mellitus (T2DM) patients with myocardial infarction (MI) and to analyze the dietary differences between patients after the first vs. subsequent cardiovascular event and between men’s and women’s diets, this study (Ms ID: ijerph-2220625) recruited 67 diabetic patients of 43-86 years old with MI and analyzed their dietary status via a questionnaire consisted of 32 questions. The study found no differences in dietary behaviors in the patients after the first vs. second MI episode. My comments are as follows:

1.     The novelty and scientific significance of this study are relatively low. This study provides no interesting novel findings. Moreover, the sample size of this study is relatively small.

2.     In Abstract: (1) the authors mention a secondary aim of this study (i.e., to analyze the differences between men’s and women’s diets), however, no results and/or conclusions about this aim are described in the Abstract; (2) the authors conclude that “they commit dietary mistakes, increasing the risk of recurrent ca don’t hesitate to get in touch with the editorial office of former MI” (lines 28-29). First, there must be an error in this sentence. Second, corresponding results will be needed to support the conclusions (I cannot see the logical relationship between the results and the conclusions in the Abstract).

3.     In the Introduction section, the authors should also introduce the rationale behind why dietary differences between patients after the first vs. subsequent CV event and between men’s and women’s diets are the aims of this study.

4.     In the Results section, physical activity, smoking, and alcohol should be considered during data analyses (put no smoking/alcohol into the exclusion criteria, or keep similar situation of smoking/alcohol/physical activity during a comparison) due to the potential strong correlation with both MI and T2DM.

5.     In the Discussion section, please first highlight the findings (particularly the new findings) of this study, and then give the discussion based on the findings. The discussion/explanation on some data, which is not relevant to the findings, should be moved to the Results section.

6.     In the Conclusions section, I recommend the authors to give some useful suggestions to readers or followers based on the findings/conclusions of this study.

7.     Typos and others: Line 38, ‘significant in women t in men’ should be ‘significant in women than in men’; line 177, ‘with an of 2.22’ should be ‘with an increase of 2.22’; lines 86-87, the ‘2’ in ‘kg/m2’ should be in superscript; line 18, ‘CV’ is first mentioned and should be given its full name (i.e., cardiovascular).

Author Response

The correction regarding reviewer comments and questions were highlighted in green. 

1. The novelty and scientific significance of this study are relatively low. This study provides no interesting novel findings. Moreover, the sample size of this study is relatively small.

Re: The authors are aware of study limitations. However, in our opinion, our results provide valuable data, and despite the small number of cases, we could reach statistical significance in many cases. In others, the p-value was high enough to expect no changes, even in larger groups.  

2. In Abstract: (1) the authors mention a secondary aim of this study (i.e., to analyze the differences between men’s and women’s diets), however, no results and/or conclusions about this aim are described in the Abstract; (2) the authors conclude that “they commit dietary mistakes, increasing the risk of recurrent ca don’t hesitate to get in touch with the editorial office of former MI” (lines 28-29). First, there must be an error in this sentence. Second, corresponding results will be needed to support the conclusions (I cannot see the logical relationship between the results and the conclusions in the Abstract).

Re:  The abstract was modified due to the suggestion

3. In the Introduction section, the authors should also introduce the rationale behind why dietary differences between patients after the first vs. subsequent CV event and between men’s and women’s diets are the aims of this study.

Re: The requested rationale was included in the "Introduction"

4. In the Results section, physical activity, smoking, and alcohol should be considered during data analyses (put no smoking/alcohol into the exclusion criteria, or keep similar situation of smoking/alcohol/physical activity during a comparison) due to the potential strong correlation with both MI and T2DM.

Re: We are aware of others' life-style health modifiers, but the current study was concentrated on diet

5. In the Discussion section, please first highlight this study's findings (particularly the new findings), and then give the discussion based on the findings. The discussion/explanation on some data, which is not relevant to the findings, should be moved to the Results section.

Re: The requested changes were introduced to the "Discussion"

6. In the Conclusions section, I recommend the authors to give some useful suggestions to readers or followers based on the findings/conclusions of this study.

Re: The  requested suggestion was added to the  "Conclusions" 

Typos and others: Line 38, ‘significant in women t in men’ should be ‘significant in women than in men’; line 177, ‘with an of 2.22’ should be ‘with an increase of 2.22’; lines 86-87, the ‘2’ in ‘kg/m2’ should be in superscript; line 18, ‘CV’ is first mentioned and should be given its full name (i.e., cardiovascular).

Re: Thank you. This was corrected.

Reviewer 3 Report

sample is small, fact that orients me for the suggestion to update in order to mention that it is a pilot study.

Abstract:

L16 - please explain an acronym when you use it for the first time (example DM, CHD, T2DM etc) and revise the entire manuscript!

L17 - I would reformulate “T2DM patients” with - patients with T2DM.

L18 – please explain “vs.”

L19 – please use patients with DM/T2DM instead of “diabetic patients”

L19 – after you explained an acronym stick to it. (myocardial infarction) and please revise the entire manuscript.

L21 – please evaluate the distribution of the patients and use standard deviation if normal or minimum and maximum if not normal.

L27-29 – please explain “ca” because the Conclusion section has lost its meaning

Introduction
L35 – the same recommendations as L16 and first L19 from the Abstract

L38 – please correct “t”

L39 – I suggest to use high blood pressure instead of “hypertension”. Please change “diabetic patients” with patients with DM and please revise the whole manuscript.

L65-67 – please reformulate more explicit the main and secondary aims (what are the differences searched – quantities, type of aliments/etc)

Material and Methods

L70-78 – please add the period of the study, please be more exact about the inclusion and exclusion criteria (the patients are consecutively admitted? etc)

L80-87 – please add the questionnaire and its validation information; please add the source for the WHO criteria for obesity/overweight

Results

L96-155 There are a lot of data reported in the Tables, which is good, but they should be trimmed and made more concise. For example, we probably do not need as many p values for all the numerous variables

Discussion

L185 – please correct if there was intended to write prevention instead of “prevent on”.

References

L244-294 - They should be updated accordingly to the journal recommendation.

L251-252, L273-274 – Please replace to a English-written source

Author Response

Thank you for your valuable comments and suggestions. The changes in the manuscript owing to your comments and suggestions are highlighted in blue.

L16 - please explain an acronym when you use it for the first time (example DM, CHD, T2DM etc) and revise the entire manuscript!

Re:  Thank you for this comment. Acronyms were explained when first appeared in the text,

L17 - I would reformulate “T2DM patients” with - patients with T2DM.

Re:  - It was done according to reviewer's suggestion

L18 – please explain “vs.”

Re: - Thank you for this comment. It was corrected

L19 – please use patients with DM/T2DM instead of “diabetic patients”

Re: - It was corrected due to the reviewer's suggestion.

L19 – after you explained an acronym stick to it. (myocardial infarction) and please revise the entire manuscript.

Re: It was done according to the reviewer's comment

L21 – please evaluate the distribution of the patients and use standard deviation if normal or minimum and maximum if not normal. 

Re: The age distribution was confirmed as normal (Gaussian). The age characteristic was presented as mean =/-SD

L27-29 – please explain “ca” because the Conclusion section has lost its meaning

Introduction

Re: Sorry for this typo. This was corrected
L35 – the same recommendations as L16 and first L19 from the Abstract

L38 – please correct “t”

Re: Sorry This was corrected

L39 – I suggest to use high blood pressure instead of “hypertension”. Please change “diabetic patients” with patients with DM and please revise the whole manuscript.

Re: Thank you. This was done.

L65-67 – please reformulate more explicit the main and secondary aims (what are the differences searched – quantities, type of aliments/etc)

Re: Aims were formulated in more exact way.

Material and Methods

L70-78 – please add the period of the study, please be more exact about the inclusion and exclusion criteria (the patients are consecutively admitted? etc)

Re: Relevant corrections were included to the text.

L80-87 – please add the questionnaire and its validation information; please add the source for the WHO criteria for obesity/overweight

The questionnaire was attached in auxiliary files (English translation). Regarding WHO criteria for obesity/overweight - they are commonly available in the web  and on  the WHO web page: https://www.who.int/news-room/fact-sheets/detail/obesity-and-overweight. If necessary we include it in the text.

Results

L96-155 There are a lot of data reported in the Tables, which is good, but they should be trimmed and made more concise. For example, we probably do not need as many p values for all the numerous variables.

Re: This subject was the matter of our internal discussion. We decided to leave it because one of the study limitations is the small number of subjects. In our opinion, the exact value in such cases is important for the reader. In the case of a small group insignificant but borderline values can be important for the reader 

Discussion

L185 – please correct if there was intended to write prevention instead of “prevent on”.

Re: Sorry for this typo. This was corrected.

References

L244-294 - They should be updated accordingly to the journal recommendation.

Re: This was corrected.

L251-252, L273-274 – Please replace to a English-written source

Re: This was replaced

Round 2

Reviewer 2 Report

1. My previous comment #1: The novelty and scientific significance of this study are relatively low. This study provides no interesting novel findings. Moreover, the sample size of this study is relatively small. To my this comment, the authors response that in their opinion the results provide valuable data, and despite the small number of cases, we could reach statistical significance in many cases. The authors did not address my comment. I believe that the authors believe that their results provide valuable data. But please show me (and show readers in the manuscript) what are the interesting/important novel findings different from previous findings?

2. My previous comment #3 and the response to this comment: In the Results section, physical activity, smoking, and alcohol should be considered during data analyses (put no smoking/alcohol into the exclusion criteria, or keep similar situation of smoking/alcohol/physical activity during a comparison) due to the potential strong correlation with both MI and T2DM. The authors did not address my comment. I know that the current study was concentrated on diet, but please consider my comment during data analyses. Otherwise, the results are questionable.

3. By the way, it seems that the authors do not like to explain how to address the comments in the point-by-point response to comments. I have to go back to the main text for the authors' answers to my comments.

Author Response

  1. My previous comment #1: The novelty and scientific significance of this study are relatively low. This study provides no interesting novel findings. Moreover, the sample size of this study is relatively small. To my this comment, the authors response that in their opinion the results provide valuable data, and despite the small number of cases, we could reach statistical significance in many cases. The authors did not address my comment. I believe that the authors believe that their results provide valuable data. But please show me (and show readers in the manuscript) what are the interesting/important novel findings different from previous findings?

Re: We are aware of the study limitations, which are declared. However, the results of our analysis are worth to be published. The new finding of this study is that high-risk people, such as patients with T2DM, are not aware of current alimentary recommendations and do not apply them even if they have experienced previous  CV events. This was highlighted in the text.

  1. My previous comment #3 and the response to this comment: In the Results section, physical activity, smoking, and alcohol should be considered during data analyses (put no smoking/alcohol into the exclusion criteria, or keep similar situation of smoking/alcohol/physical activity during a comparison) due to the potential strong correlation with both MI and T2DM. The authors did not address my comment. I know that the current study was concentrated on diet, but please consider my comment during data analyses. Otherwise, the results are questionable.

Re: Thank you for this suggestion. We know that complex lifestyle modification is necessary to reduce the risk of CV disorders. However, to maintain the study’s integrity, we not included mentioned by reviewer questions in the analysis and concentrated on the diet. This was also declared as a study limitation.

  1. By the way, it seems that the authors do not like to explain how to address the comments in the point-by-point response to comments. I have to go back to the main text for the authors' answers to my comments.

Re: We are sorry for this. The current corrections were visibly included in the text in tracking changes mode.